# Changes in Japanese physicians' relationships with the pharmaceutical industry between 2008 and 2021: A national survey

**Sayaka Saito[1], Kei Mukohara[2]\*, Kazuhiro Shimomura[3,4], Kenta Murotani[5]**

**1** Department of General Medicine, National Hospital Organization Kasumigaura Medical Center, Tsuchiura, Ibaraki, Japan, **2** Department of General and Family Medicine, Kurume University Medical Center, Kurume, Fukuoka, Japan, **3** Department of Biostatistics, Graduate School of Medicine, Kurume University, Kurume, Fukuoka, Japan, **4** Department of Pharmacy, Aichi Cancer Center Hospital, Nagoya, Aichi, Japan, **5** Biostatistics Center, Kurume University, Kurume, Fukuoka, Japan

\* mukohara_kei@kurume-u.ac.jp

**Data Availability Statement:** All relevant data are within the manuscript and its Supporting Information files.

## Abstract

### Background

A national survey we conducted in 2008 showed that many Japanese physicians interacted with and received gifts from pharmaceutical representatives (PRs) and had a positive attitude toward relationships with PRs. The revised promotion code of the Japan Pharmaceutical Manufacturers Association in 2019 prohibited the provision of non-educational promotional aids including sticky notes, mouse pads, and calendars. During the COVID-19 pandemic in 2020, face-to-face meetings were socially restricted. This study assessed the extent of current Japanese physicians' involvement in pharmaceutical promotional activities and their attitudes toward relationships with PRs and to ascertain any changes between 2008 and 2021. We also examined the factors that predicted positive attitudes toward gifts from PRs.

### Methods

From January to March 2021, we conducted a national mail survey of Japanese physicians in seven specialties: internal medicine, surgery, orthopedics, pediatrics, obstetrics-gynecology, psychiatry, and ophthalmology.

### Results

There were 1636 participants and the response rate was 63.2%. Most physicians met face-to-face with PRs (78.8%), whereas only a minority received meals outside the workplace (4.5%). PRs were thought to have an important role in continuing medical education (66.1%) and to provide accurate information about new drugs (74.2%). Opinions were divided on the appropriateness of gifts from PRs. Most thought that stationery and meals provided by the industry did not affect prescribing behavior (89.7% and 75.8%, respectively). Factors that predicted a positive attitude toward gifts from PRs were male, orthopedic

**Funding:** This work was supported by JSPS KAKENHI Grant Numbers JP 20H03920 and JP 22H04923 (CoBiA).

**Competing interests:** The authors have declared that no competing interests exist.

specialty vs. internal medicine, more interactions with PRs, a positive attitude toward informational value, and no rules banning meetings with PRs.

## Conclusion

Involvement in pharmaceutical promotional activities is still common among practicing physicians in Japan, although the extent of the involvement had declined. Rules banning meetings with PRs appear to continue being effective at limiting a physician's involvement with promotional activities and their critical attitudes toward gifts from PRs.

## Introduction

Involvement in pharmaceutical promotional activities is widespread among physicians in many countries [1–4]. Studies have shown that receiving gifts from pharmaceutical representatives (PRs) is associated with inappropriate prescribing behavior that undermines patient interests, which is a common concern worldwide [5–7]. In the 2000s, governmental, nongovernmental, and industry self-imposed regulations on pharmaceutical promotion were implemented in some countries [8]. In the United States, for example, the Physician Payment Sunshine Act was enacted in 2010, which required the reporting of all pharmaceutical industry payments to physicians [9]. Around the same time, policy proposals to control conflicts of interest were issued by a group of academic leaders, the Association of American Medical Colleges, and the Institute of Medicine [10–12]. In 2009, a self-regulatory code of the pharmaceutical industry was revised to prohibit many forms of gifts that are not considered educational [13]. After these regulatory changes, the extent of promotional activities began to decline [14–17].

In Japan, a national survey we conducted in 2008 showed that many physicians interacted with and received gifts from PRs [18]. It also found that many physicians valued information from PRs and thought they were not influenced by discussions with PRs or by receiving gifts. Since then, there have been few new regulations set by the Japanese government or policy proposals from professional organizations. However, the Japan Pharmaceutical Manufacturers Association adopted a transparency self-regulatory code similar to that in Europe [8, 19], which was revised following the International Federation of Pharmaceutical Manufacturers and Associations code revision [20]. The latest code revision in 2019 prohibited the provision of non-educational promotional aids such as sticky notes, mouse pads, and calendars. During the COVID-19 pandemic in 2020, face-to-face meetings were socially restricted. This study aimed to assess the extent of current Japanese physicians' involvement in pharmaceutical promotional activities and their attitudes toward relationships with PRs and to ascertain whether there have been any changes between 2008 and 2021. We also examined which factors predicted positive attitudes toward gifts from PRs.

## Methods

### Ethics

The Ethical Committee of Kurume University (health care ethics) approved the study protocol (study number 20096). At the beginning of the questionnaire, the respondents needed to answer "agree" to a question asking whether they consented to participate in the study.

## Study design and participants

The study was a cross-sectional mail survey of Japanese practicing physicians. The target population was practicing physicians working in clinical and hospital settings in Japan. As in the survey in 2008, we selected four clinic-based and four hospital-based physicians in each of seven specialties (internal medicine, general surgery, orthopedics, pediatrics, obstetrics-gynecology, psychiatry, and ophthalmology) for all 47 Japanese prefectures [18]. Because there is no complete updated registry of Japanese physicians, we used lists of medical institutions from the Regional Bureaus of Health and Welfare to randomly select clinics and hospitals. Then, we identified survey participants on the websites of these institutions. We excluded those who worked in academic medical centers, were retired or on leave, or were in administrative positions in hospitals.

## Survey administration

The survey was administered from January to March 2021. We used the same administration methods as for the survey in 2008 [18]. A pre-notification postcard was sent to all participants. One week after sending the pre-notification postcard, we sent every participant a cover letter, questionnaire, self-addressed postcard with the individual participant's name, stamped reply envelope, and a pen (worth approximately US $3) as an incentive. The cover letter stated the purpose of the study, the voluntary nature of participation, and the confidentiality of the responses. We asked participants to post the completed questionnaire separately from the self-addressed postcard, which allowed us to track respondents whilst maintaining their anonymity. Non-respondents received up to other two reminders along with copies of the original questionnaire at two-week intervals.

## Survey instrument

The survey was a 28-item, four-page, anonymous, self-administered questionnaire (S1 File). The questionnaire was developed based on a discussion between two authors (SS, KM) considering the previous findings in 2008 and changes in social situations. It included 13 items from 23 items in the 2008 survey and 15 items were modified or newly added [18]. It asked about background information of the respondents, pharmaceutical promotions at the workplace, frequency of involvement in promotional activities, and attitudes toward relationships with PRs. The questionnaire was not pilot tested before the study administration.

Background information of respondents included sex, age, year of graduation from medical school, specialty, and practice setting. Since previous studies have indicated that physicians who prescribe more also receive more gifts, we compared the differences by specialty and expected that internists, who are known to prescribe more, would be more involved in pharmaceutical promotional activities than other specialists. We also asked whether their workplaces had rules banning meetings with PRs, gifts, or both. We added a new question asking whether the rules, if any, aimed to prevent the spread of COVID-19 infection. We also asked three questions about promotional items or activities at their workplaces: usage of items (e.g., pens) bearing the name of a pharmaceutical company; placement of items (e.g., calendars) with the name of a pharmaceutical company in examination rooms or waiting rooms; and visits from PRs.

We asked how often they were involved in each of eight promotional activities in the same manner as the 2008 survey using a six-point Likert scale: never, less than once a month, two to three times a month, once a week, two to three times a week, and nearly every day. Three of the eight activities were the same as in the 2008 survey: face-to-face meetings with PRs; receiving stationery; and receiving meals outside the workplace. Five newly added activities were

online meetings with PRs, e-mail communications with PRs, receiving medical textbooks, participating in a promotional meeting with food at the workplace, and participating in online educational lectures in which the pharmaceutical industry is involved.

We asked about their degree of agreement with statements about relationships with PRs using a 5-point Likert scale: agree, somewhat agree, neutral, somewhat disagree, and disagree. Three statements about the informational values of PRs were identical to those in the 2008 survey: PRs have an important role in continuing medical education (CME) for practicing physicians; PRs provide accurate information on new medications; and PRs provide accurate information on established medications ("informational value"). Four statements about appropriateness were modified from the 2008 survey: it is ethically appropriate to receive stationary, medical textbooks, and meals outside the workplace, and to participate in a promotional meeting with food in the workplace ("appropriateness"). Among statements about the influence of relationships with PRs ("influence"), one was identical to that in the 2008 survey: their prescribing behavior was influenced by discussions with PRs. The other two statements were newly added: their behavior was influenced by receiving stationery and their behavior was influenced by food or meals.

## Statistical analysis

A questionnaire was considered to be evaluable when it was returned by the pre-specified deadline (March 9, 2021), had an agreement to the first question asking for consent to participate, and had complete information for 50% or more of all 28 questions except the last one. Multiple responses to a question were considered no answer. SAS9.4 (SAS Institute Inc., Cary, NC) was used for all statistical analyses.

To compare the results of this survey with the 2008 results, ordinal logistic regression analysis was used and $P$-values for the survey year were described. Dependent variables were three items of involvement (meeting with PRs, stationery, and meals), the informational value of PRs, and the influence of PRs, which were also asked for in the 2008 survey. Independent variables were the survey year (2008 or 2021), sex, and clinical settings (clinic or hospital).

Multiple linear regression analysis was conducted to examine factors that predicted the frequency of physicians' interactions with PRs. Dependent variables were the sum of the responses to three questions: face-to-face meetings with PRs, online meetings with PRs, and e-mail communication with PRs (not at all = 0, less than once a month or more = 1; range = 0–3). Independent variables were sex, age, working environment (clinic = reference), specialty (internal medicine = reference), and presence of rules (no rules = reference).

Multiple regression analysis (forced entry method) was also conducted to examine the factors that predicted a positive attitude toward gifts from PRs. Dependent variables were the sum of the responses to four "appropriateness" questions (inappropriate/not very appropriate/ neutral = 0, fairly appropriate/appropriate = 1; range = 0–4). Independent variables included sex, age, working environment (clinic = reference), specialty (internal medicine = reference), presence of rules (no rules = reference), sum of the responses to eight questions regarding involvement in promotional activities (not at all = 0, less than once a month or more = 1; range = 0–8), sum of the responses to three statements regarding "informational value" (agree/ somewhat agree = 1 and neutral/somewhat disagree/disagree = 0: range 0–3), and the sum of the responses to three statements regarding "influence" (not influential at all/not very influential = 1 and influential/somewhat influential/neutral = 0; range 0–3).

For all statistical analyses, $P$-values were 2-tailed and those less than 0.05 were considered statistically significant.

## Results

### Characteristics of respondents

Of the 2632 people sampled, 44 were ineligible and 163 declined to participate. Of the 2588 judged to be eligible, 1636 completed the survey, a response rate of 63.2%. Table 1 shows the characteristics of the survey respondents in 2021 and 2008. Males accounted for 80.8% of respondents, which was similar to the 2018 national physician statistics [21]. The average age of the respondents was 56.7 years (range, 27–94 years, standard deviation 12.8), and 51.4% had graduated ≥31 years previously, which is higher than the national statistics [21]. Compared with the 2008 survey, there were significantly fewer physicians that had graduated ≤20 years previously, and more that had graduated ≥3 years previously. The number of physicians working in hospitals (52.1%) and clinics (47.9%) was similar between surveys. Regarding the rules banning PR visits and/or gifts at the workplaces, 5.4% of responses indicated there were rules banning PR visits and gifts, 10.0% indicated there were rules banning PR visits but not gifts,

**Table 1. Characteristics of survey respondents in 2021 and 2008.**

| | | Survey year | | | |
|---|---|---|---|---|---|
| | | 2021 (n = 1636) | | 2008 (n = 1411) | |
| Characteristic | | No. | (%) | No. | (%) |
| Sex | | | | | |
| Male | | 1314 | (80.8) | 1084 | (76.9) |
| Female | | 313 | (19.2) | 326 | (23.1) |
| No. of years in practice | | | | | |
| ≤10 | | 144 | (8.9) | 339 | (24.0) |
| 11–20 | | 240 | (15.0) | 488 | (34.6) |
| 21–30 | | 393 | (24.6) | 428 | (30.3) |
| ≥31 | | 818 | (51.4) | 155 | (11.0) |
| Specialty | | | | | |
| Internal medicine | | 243 | (14.9) | 214 | (15.3) |
| General surgery | | 185 | (11.3) | 181 | (12.8) |
| Orthopedic surgery | | 213 | (13.0) | 177 | (12.5) |
| Pediatrics | | 262 | (16.0) | 221 | (15.7) |
| Obstetrics/Gynecology | | 236 | (14.4) | 210 | (14.9) |
| Psychiatry | | 192 | (11.7) | 197 | (14.0) |
| Ophthalmology | | 243 | (14.9) | 209 | (14.8) |
| Practice setting | | | | | |
| Clinic | | 828 | (52.1) | 588 | (41.7) |
| Hospital | | | | 822 | (58.3) |
| Private hospital | | 526 | (33.1) | | |
| Public hospital | | 254 | (15.5) | | |
| Other | | 9 | (0.6) | | |
| Rules banning meetings with PRs and/or gifts at the workplace | | | | | |
| Yes | Rules banning meetings and gifts | 88 | (5.4) | 63 | (4.5) |
| | Rules banning meetings, not gifts | 164 | (10.0) | 54 | (3.9) |
| | Rules banning gifts, not meetings | 106 | (6.5) | 217 | (15.6) |
| No | | 1002 | (61.2) | 1057 | (76.1) |
| Don't know | | 262 | (16.0) | n/a | |

Abbreviation: PR, pharmaceutical representative. n/a, not available.

6.5% reported there were rules banning gifts but not PR visits, and 22.1% reported some rules, which was similar to the 2008 survey.

## Promotions at the workplaces

Regarding promotional items and activities in the work environment, 61.9% of respondents used items (e.g., pens) bearing a company name in an examination room, 51.0% reported placement of items (e.g., a calendar) with a company name in examination rooms and/or waiting spaces, and 89.1% reported visits from PRs at their workplaces.

## Involvement in promotional activities

Table 2 shows physician involvement in promotional activities. Most respondents (87.0%) had interactions with PRs through face-to-face meetings, online meetings, or e-mail communications Stationary was accepted by approximately one-fourth of physicians (25.7%) and medical textbooks were accepted by less than one-tenth of physicians (8.7%). Many physicians attended web lectures promoted by the pharmaceutical industry (75.5%) and fewer attended promotional meetings with food provided by PRs at the workplace (38.3%) or received meals outside the workplace (4.5%).

## Attitudes toward relationships with PRs

The results of an analysis of attitudes toward relationships with PRs are shown in Fig 1. Many respondents thought that PRs have an important role in CME (66.1%) and that they provide accurate information about new drugs (74.2%). Opinions were divided on the appropriateness of accepting stationery and medical textbooks, as well as attending promotional meetings at their workplace. Two-thirds of respondents felt that accepting food from PRs was not appropriate (68.2%). However, most felt that accepting stationery (89.7%) and meals (75.8%) provided by the industry did not affect their prescribing behavior.

## Comparison between the 2021 and 2008 surveys

Percentages of physicians with interactions, including meetings, with PRs and accepting stationery and meals were lower than those in 2008 (Tables 2 and S1).

Compared with the 2008 survey, fewer physicians felt that PRs have an important role in CME, whereas more physicians felt that interactions with PRs had an impact on their prescribing behavior (Fig 1 and S2 Table). Differences in the percentage of physicians who felt that PRs provide accurate information about new or established medications were not significantly different.

## Multivariate predictors of interactions with PRs (Table 3)

Factors that predicted interactions with PRs (face-to-face meetings, online meetings, and e-mail communications) included physician specialty (psychiatry vs. internal medicine, and ophthalmology vs. internal medicine) and presence of rules (rules banning meetings and gifts vs. no rules and not knowing vs. no rules).

## Multivariate predictors of positive attitudes toward gifts from PRs

Factors that predicted positive attitudes toward gifts from PRs (i.e. physicians thought gifts from PRs were appropriate) were male vs. females, specialty of orthopedics vs. internal medicine, more interactions with PRs, positive attitude toward informational value, no rules vs. rules banning meetings and gifts or rules banning meetings but allowing gifts (Table 4).

**Table 2. Physician involvement in various types of pharmaceutical promotional activities.**

| Type of promotional activities | Survey year | | | | P-value† |
| --- | --- | --- | --- | --- | --- |
| | 2021 (n = 1636) | | 2008 (n = 1411) | | |
| | No. | (%) | No. | (%) | |
| Interactions with PRs | | | | | |
| Face-to-face meetings with PRs | 1280 | (78.8) | 1383 | (98.3) | < 0.001 |
| Online meetings with PRs | 399 | (24.4) | | | |
| E-mail communications with PRs | 859 | (52.6) | | | |
| Any of the above | 1423 | (87.0) | | | |
| Stationery | 419 | (25.7) | 1347 | (95.7) | < 0.001 |
| Medical textbook | 132 | (8.1) | | | |
| Industry-sponsored web seminars | 1235 | (75.5) | | | |
| Promotional meetings at workplaces | 625 | (38.3) | | | |
| Meals outside the workplaces | 73 | (4.5) | 697 | (49.4) | < 0.001 |

†Pearson's chi-square test.

Abbreviation: PR, pharmaceutical representative.

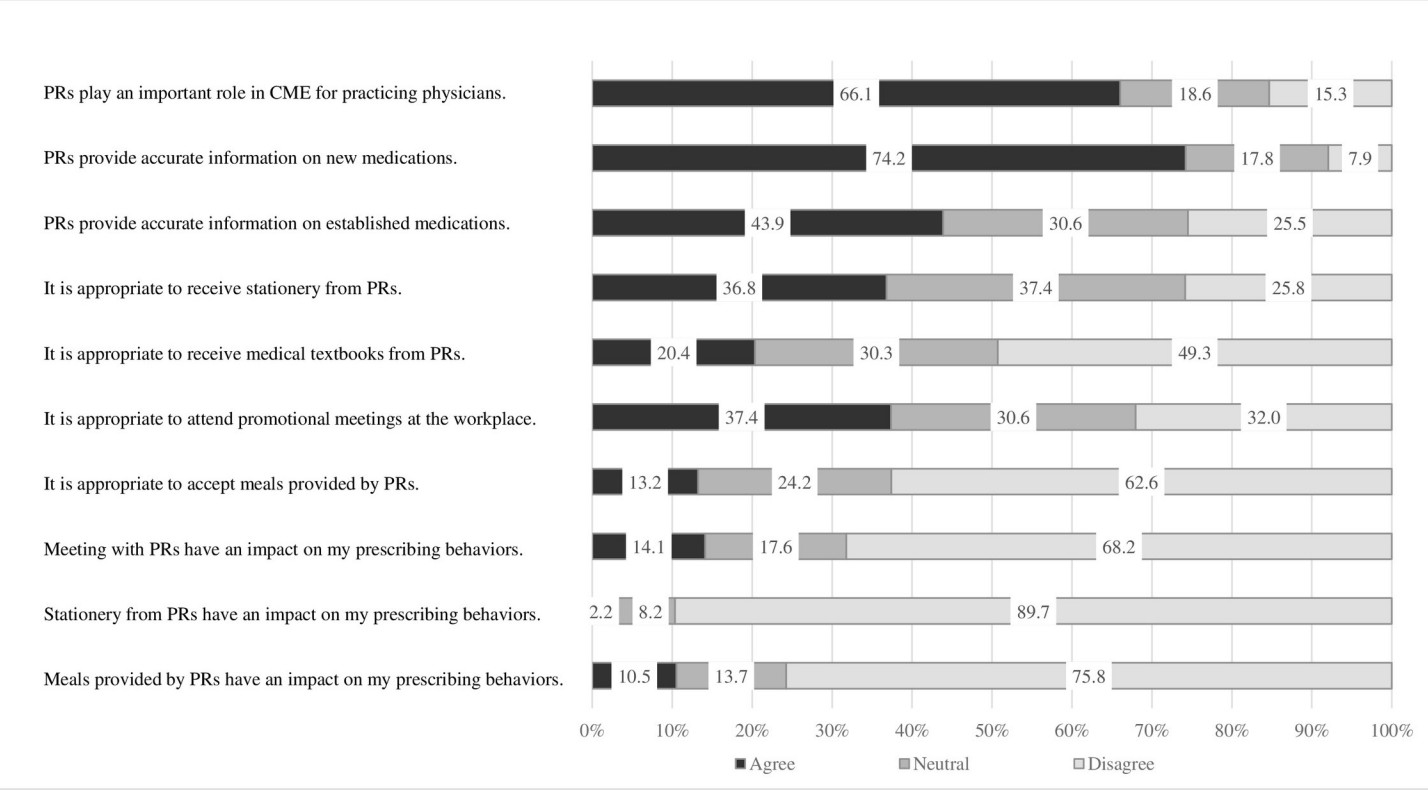

**Fig 1. Attitudes toward relationships with pharmaceutical representatives.** Abbreviation: PR, pharmaceutical representative.

**Table 3. Multivariate predictors of interactions with PRs.**

| Independent variables | | Coefficient | 95% confidence interval | *P*-value |
|---|---|---|---|---|
| Sex | Male (= 1) vs. female (= 0) | 0.06 | −0.07 to 0.18 | 0.35 |
| Age | | 0.00 | −0.01< to <0.01 | 0.19 |
| Clinical setting | Private hospital (= 1) vs. clinics (= 0) | 0.02 | −0.14 to 0.09 | 0.69 |
| | Public hospital (= 1) vs. clinics (= 0) | 0.04 | −0.11 to 0.20 | 0.59 |
| | Others (= 1) vs. clinics (= 0) | −0.008 | −0.60 to 0.58 | 0.98 |
| Specialty | Surgery (= 1) vs. Internal medicine (= 0) | −0.12 | −0.30 to 0.06 | 0.19 |
| | Orthopedics (= 1) vs. Internal medicine (= 0) | 0.01 | −0.17 to 0.19 | 0.92 |
| | Pediatrics (= 1) vs. Internal medicine (= 0) | 0.13 | −0.04 to 0.30 | 0.13 |
| | Obstetrics-gynecology (= 1) vs. Internal medicine (= 0) | −0.05 | −0.22 to 0.12 | 0.57 |
| | Psychiatry (= 1) vs. Internal medicine (= 0) | 0.21 | 0.03 to 0.39 | 0.02 |
| | Ophthalmology (= 1) vs. Internal medicine (= 0) | 0.20 | 0.03 to 0.37 | 0.02 |
| Rules at workplace | Banning meetings and gifts (= 1) vs. no rules (= 0) | −0.29 | −0.51 to −0.07 | <0.01 |
| | Banning gifts (= 1) vs. no rules (= 0) | −0.04 | −0.21 to 0.13 | 0.64 |
| | Banning meetings (= 1) vs. no rules (= 0) | −0.17 | −0.37 to 0.03 | 0.09 |
| | Not knowing (= 1) vs. no rules (= 0) | −0.22 | −0.36 to −0.07 | <0.01 |

Dependent variables included the sum of the responses to three question item: face-to-face meetings with PRs, online meetings with PRs, and e-mail communication with PRs (not at all = 0, less than once a month or more = 1; range = 0–3).

## Discussion

The involvement in promotional activities among Japanese physicians was decreased significantly in 2021 compared with that in 2008, whereas rules banning gifts from PRs did not

**Table 4. Multivariate predictors of physicians' positive attitudes toward gifts from PRs.**

| Independent variables | | Coefficient | 95% confidence interval | *P*-value |
|---|---|---|---|---|
| Informational value (least evaluating = 0; range0–3) | | 0.25 | 0.18 to 0.31 | <0.01 |
| Influence (perceiving influence = 0; range0–3) | | −0.01 | −0.09 to 0.07 | 0.78 |
| Physician-industry relationships (no relationship = 0; range 0–8) | | 0.15 | 0.10 to 0.19 | <0.01 |
| Sex | Male (= 1) vs. female (= 0) | 0.21 | 0.003 to 0.41 | 0.046 |
| Age | | 0.00 | −0.01< to <0.01 | 0.31 |
| Clinical setting | Private hospital (= 1) vs. clinics (= 0) | 0.15 | −0.04 to 0.34 | 0.11 |
| | Public hospital (= 1) vs. clinics (= 0) | −0.05 | −0.30 to 0.20 | 0.70 |
| | Other (= 1) vs. clinics (= 0) | −0.67 | −1.61 to 0.28 | 0.16 |
| Specialty | Surgery (= 1) vs. internal medicine (= 0) | −0.02 | −0.32 to 0.27 | 0.87 |
| | Orthopedics (= 1) vs. internal medicine (= 0) | 0.28 | 0.001 to 0.56 | 0.049 |
| | Pediatrics (= 1) vs. internal medicine (= 0) | −0.09 | −0.36 to 0.18 | 0.50 |
| | Obstetrics-gynecology (= 1) vs. internal medicine (= 0) | 0.14 | −0.14 to 0.41 | 0.33 |
| | Psychiatry (= 1) vs. internal medicine (= 0) | −0.14 | −0.43 to 0.16 | 0.36 |
| | Ophthalmology (= 1) vs. internal medicine (= 0) | −0.13 | −0.41 to 0.15 | 0.36 |
| Rules at workplace | Banning meetings and gifts (= 1) vs. no rules (= 0) | −0.41 | −0.76 to −0.05 | 0.02 |
| | Banning gifts, not meetings (= 1) vs. no rules (= 0) | −0.20 | −0.47 to 0.07 | 0.14 |
| | Banning meetings, not gifts (= 1) vs. no rules (= 0) | 0.36 | 0.04 to 0.68 | 0.03 |
| | Not knowing (= 1) vs. no rules (= 0) | −0.13 | −0.37 to 0.11 | 0.28 |

Dependent variables included the sum of the responses to four "appropriateness" questions (inappropriate/not very appropriate/neutral = 0, fairly appropriate/appropriate = 1; range = 0–4).

increase. There were small yet statistically significant changes in the attitudes of Japanese physicians in terms of the informational value of PRs for CME and influence of gifts from PRs.

Previous studies reported that interactions between physicians or medical students and pharmaceutical companies became less frequent in the U.S. [14, 16, 22] where rules to limit these interactions became more common in residency programs and medical schools [12, 17]. This discipline to limit physicians' behavior is thought to have directly contributed to a decrease in their involvement in promotional activities. In the current survey, rules banning meetings with PRs were increased but those banning gifts were not. We speculated that a significant decrease in face-to-face meetings with PRs and accepting meals outside the workplace was not due to a change in physicians' ethical concerns about relationships with PRs, but more as a result of rules that do not allow PRs to enter their workplaces.

Of note, those who did not know whether their workplace had rules banning meetings or gifts were less likely to interact with PRs than those in workplaces that had no rules. The reason for this finding is not clear but might be that those who were unaware of the rules did not care about interacting with PRs

Previous studies examining chronological changes in physicians' attitudes toward promotional activities are limited. In a survey of U.S. medical students, the number of medical students who thought grand rounds were educational and helpful decreased from 89.0% (2003) to 67.0% (2012). We found that the percentage of respondents who thought that the pharmaceutical industry has an important role in CME decreased from 73.5% in 2008 to 66.1% in 2021, although the degree of change was smaller than that in a U.S. student survey.

In our 2008 survey, when asked about the appropriateness of these relationships, 37% and 85% of respondents agreed that gifts of low- and high-monetary value were ethically inappropriate, respectively. In this survey, 25.8% and 62.6% of the respondents said that it was not appropriate to receive stationery and meals, respectively. It is difficult to compare these findings because the questions were asked differently, but it appears that the awareness of ethical issues about receiving gifts has not increased.

There was a statistically significant increase from 5.7% to 14.1% in the number of respondents who thought that meeting with PRs would influence their prescribing behavior; however, there was little change in the number of respondents who thought it would not (68.9% in 2008, 68.3% in 2021).

Although positive attitudes toward gifts from PRs were predicted by more interactions with PRs and positive attitudes toward informational values as well as rules banning meetings, attitudes toward influence from PRs were not. This finding indicates that limiting involvement in promotional activities rather than emphasizing the negative influence of gifts from the pharmaceutical industry might effectively facilitate physicians to think critically about their relationships with PRs.

There were several study limitations. First, because this was not a follow-up survey of the same population in the 2008 survey, the differences between the survey results cannot be directly compared. Second, unlike similar studies conducted in the US, there is no legislation or professional organization/medical school decision in Japan that discourages physicians from interacting with or accepting gifts and promotional materials from pharmaceutical representatives. Therefore, the variable studied between 2008 and now is not a determining factor to change attitudes or behaviors, as per the study findings. Third, although the survey was anonymous, the answers may have been biased toward social desirability Fourth, the cross-sectional nature of this study does not allow us to infer the causality of the associations.

Involvement in pharmaceutical promotional activities is still common among practicing physicians in Japan, although the extent of the involvement had declined. Rules banning

meetings with PRs appear to continue being effective at limiting a physician's involvement with promotional activities and their critical attitudes toward gifts from PRs.

## Supporting information

**S1 File. The survey instrument in English.**
(DOCX)

**S1 Table. Physician-industry relationships compared with the 2008 survey.** Abbreviation: PR, pharmaceutical representative. †Multivariable ordinal logistic regression adjusted for the survey year (2008 or 2021), sex, and clinical setting (clinic or hospital).
(DOCX)

**S2 Table. Physicians' attitudes compared with the 2008 survey.** Abbreviation: PR, pharmaceutical representative. †Multivariable ordinal logistic regression adjusted for the survey year (2008 or 2021), sex, and clinical setting (clinic or hospital).
(DOCX)

## Acknowledgments

We would like to thank Yukiko Tanaka and Madoka Tsutsumi, MPH, from Wintria Inc. for their assistance in preparing, distributing, and collecting survey forms, as well as for data entry. We would like to thank J. Ludovic Croxford, PhD, from Edanz (https://jp.edanz.com/ac) for editing a draft of this manuscript.

## Author Contributions

**Conceptualization:** Sayaka Saito, Kei Mukohara.

**Data curation:** Sayaka Saito.

**Formal analysis:** Kazuhiro Shimomura, Kenta Murotani.

**Funding acquisition:** Kei Mukohara.

**Methodology:** Sayaka Saito, Kei Mukohara.

**Supervision:** Kei Mukohara.

**Writing – original draft:** Sayaka Saito.

**Writing – review & editing:** Kei Mukohara, Kazuhiro Shimomura, Kenta Murotani.

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
