## [Decision Letter · Decision Letter 0]

6 Mar 2023

PONE-D-22-31149Changes in Japanese physicians’ relationships with the pharmaceutical industry between 2008 and 2021: a national surveyPLOS ONE

Dear Dr. Mukohara,

Thank you for submitting your manuscript to PLOS ONE. After careful consideration, we feel that it has merit but does not fully meet PLOS ONE’s publication criteria as it currently stands. Therefore, we invite you to submit a revised version of the manuscript that addresses the points raised during the review process.

We look forward to receiving your revised manuscript.

Kind regards,

Soham Bandyopadhyay

Academic Editor

PLOS ONE

Journal Requirements:

3. PLOS requires an ORCID iD for the corresponding author in Editorial Manager on papers submitted after December 6th, 2016. Please ensure that you have an ORCID iD and that it is validated in Editorial Manager. To do this, go to ‘Update my Information’ (in the upper left-hand corner of the main menu), and click on the Fetch/Validate link next to the ORCID field. This will take you to the ORCID site and allow you to create a new iD or authenticate a pre-existing iD in Editorial Manager. Please see the following video for instructions on linking an ORCID iD to your Editorial Manager account: https://www.youtube.com/watch?v=_xcclfuvtxQ.

Reviewers' comments:

Reviewer's Responses to Questions

**Comments to the Author**

1. Is the manuscript technically sound, and do the data support the conclusions?

Reviewer #1: Yes

Reviewer #2: Partly

2. Has the statistical analysis been performed appropriately and rigorously? 

Reviewer #1: Yes

Reviewer #2: I Don't Know

3. Have the authors made all data underlying the findings in their manuscript fully available?

Reviewer #1: Yes

Reviewer #2: No

4. Is the manuscript presented in an intelligible fashion and written in standard English?

Reviewer #1: Yes

Reviewer #2: Yes

5. Review Comments to the Author

Reviewer #1: i think its important think about the Pharmaceutical sales. Pharmaceutical representatives doing a good job to promote their companies product. How they can convince the doctor to increase the sale of their product is important. I am excited that the japanese specialty doctors are not influenced by the promotions given by the PRs. However, the study does not seem novel.

Reviewer #2: The article discusses the involvement of physician in pharmaceutical promotional activities and attempts to track the changes from a previous 2008 study. They used a survey tool that partly resembles their previous tool which makes it hard to track this change. Also the survey tool was not part of the publication, not knowing the language of the question makes it hard to understand the responses provided. There was no logical explanation on why different specialties was a determining factor for significance. In addition, unlike the research done on sillier topics in the US, there is no legislation or decision from professional organizations or medical schools that deters physician from interacting/accepting gifts and promotional material from pharmaceutical representative, so the variable studied between 2008 and now is, and based on the study findings, not a determining factor to change attitudes/behaviors.

6. PLOS authors have the option to publish the peer review history of their article (what does this mean?). If published, this will include your full peer review and any attached files.

Reviewer #1: No

Reviewer #2: No

---

## [Author Response · Author response to Decision Letter 0]

25 Apr 2023

Reviewer #1: I think its important think about the pharmaceutical sales. Pharmaceutical representatives doing a good job to promote their companies product. How they can convince the doctor to increase the sale of their product is important. I am excited that the Japanese specialty doctors are not influenced by the promotions given by the PRs. However, the study does not seem novel.

Response: Thank you for your review. As you noted, there have been numerous studies conducted on this topic in the past. However, this particular study is significant as it sheds light on the fact that the perception of Japanese physicians has not changed over time, despite long-standing concerns. We believe that this finding is a valuable contribution to the existing body of knowledge on physician-pharmaceutical industry relationships.

Reviewer #2: The article discusses the involvement of physician in pharmaceutical promotional activities and attempts to track the changes from a previous 2008 study. They used a survey tool that partly resembles their previous tool which makes it hard to track this change. Also the survey tool was not part of the publication, not knowing the language of the question makes it hard to understand the responses provided. 

Response: Thank you for your comment. We have added a survey form in which the questions are in English as a supplementary file.

Reviewer #2: There was no logical explanation on why different specialties was a determining factor for significance. 

Response: Since previous studies have indicated that physicians who prescribe more also receive more gifts, we compared the differences by specialty and expected that internists, who are known to prescribe more, would be more involved in pharmaceutical promotional activities than other specialists. We have added an explanation of this point on page 7, line 19 to page 8, line 3 of the manuscript.

Reviewer #2: In addition, unlike the research done on similar topics in the US, there is no legislation or decision from professional organizations or medical schools that deters physician from interacting/accepting gifts and promotional material from pharmaceutical representative, so the variable studied between 2008 and now is, and based on the study findings, not a determining factor to change attitudes/behaviors.

Response: We agree that the variable studied between 2008 and now is not a determining factor to change attitudes or behaviors, as per the study findings. We have added an explanation of this point on page 25 lines 3 to 7 of the manuscript.

---

## [Editor Report · Decision Letter 1]

15 May 2023

Changes in Japanese physicians’ relationships with the pharmaceutical industry between 2008 and 2021: a national survey

PONE-D-22-31149R1

Dear Dr. Mukhora,

We’re pleased to inform you that your manuscript has been judged scientifically suitable for publication and will be formally accepted for publication once it meets all outstanding technical requirements.

Kind regards,

Soham Bandyopadhyay

Academic Editor

PLOS ONE

---

## [Editor Report · Acceptance letter]

22 May 2023

PONE-D-22-31149R1 

Changes in Japanese physicians’ relationships with the pharmaceutical industry between 2008 and 2021: a national survey 

Dear Dr. Mukohara:

I'm pleased to inform you that your manuscript has been deemed suitable for publication in PLOS ONE. Congratulations! Your manuscript is now with our production department. 

Kind regards, 

on behalf of

Dr. Soham Bandyopadhyay 

Academic Editor

PLOS ONE